# A Review of Viruses Infecting Yam (*Dioscorea* spp.)

**DOI:** 10.3390/v14040662

**Published:** 2022-03-23

**Authors:** Guo-Fu Luo, Ana Podolyan, Dawit B. Kidanemariam, Carmel Pilotti, Gary Houliston, Amit C. Sukal

**Affiliations:** 1Centre for Pacific Crops and Trees (CePaCT), Land Resources Division, Pacific Community (SPC), Suva, Fiji; guoful@spc.int (G.-F.L.); carmelp@spc.int (C.P.); 2Manaaki Whenua—Landcare Research, Lincoln 7640, New Zealand; podolyana@landcareresearch.co.nz (A.P.); houlistong@landcareresearch.co.nz (G.H.); 3Department of Plant and Soil Sciences, Forestry and Agricultural Biotechnology Institute (FABI), University of Pretoria, Pretoria 0002, South Africa; dawiteth@gmail.com

**Keywords:** yam viruses, potyvirus, yam mosaic virus, yam mild mosaic virus, badnavirus, macluravirus, *Dioscorea* spp.

## Abstract

Yam is an important food staple for millions of people globally, particularly those in the developing countries of West Africa and the Pacific Islands. To sustain the growing population, yam production must be increased amidst the many biotic and abiotic stresses. Plant viruses are among the most detrimental of plant pathogens and have caused great losses of crop yield and quality, including those of yam. Knowledge and understanding of virus biology and ecology are important for the development of diagnostic tools and disease management strategies to combat the spread of yam-infecting viruses. This review aims to highlight current knowledge on key yam-infecting viruses by examining their characteristics, genetic diversity, disease symptoms, diagnostics, and elimination to provide a synopsis for consideration in developing diagnostic strategy and disease management for yam.

## 1. Introduction

Yam (*Dioscorea* spp.) is the collective name for a group of multi-species, dioecious, and monocotyledonous vines cultivated primarily for their starchy underground tubers. The *Dioscorea* genus belongs to the family *Dioscoreaceae* of the order *Dioscoreales* and comprises approximately 600 species of domesticated and wild yams. However, only about 10 edible species are widely grown for food in the tropical and sub-tropical regions of the world [1,2,3,4,5]. In particular, yam is a major food staple and an important source of carbohydrates for the people of Africa, Central and South America, parts of Asia, the Caribbean, and the Pacific Islands [6,7,8,9,10]. In addition, yam holds significant economic, social and cultural value wherever it is cultivated [11,12,13].

Globally, yam is the fourth most important root and tuber crop by production after potato, sweet potato, and cassava. The ‘yam belt’ of West Africa, comprising Benin, Ivory Coast, Ghana, Nigeria, and Togo, contributes about 95% towards global yam production [14]. Despite its socio-economic importance, yam production is heavily constrained by many factors, such as the high cost and availability of clean “seeds”, pests, and diseases [2,15,16]. Pests and diseases, such as bacteria, fungi, insects, nematodes, and viruses, have direct negative impacts on yield and quality. Of these, viruses pose the most serious problems as they are the hardest to control, easily spread with planting material, and have been reported in all yam growing regions of the world [2,6,16,17].

Viruses belonging to the families *Alphaflexiviridae* (genus *Potexvirus*), *Betaflexiviridae*, *Bromoviridae* (genus *Cucumovirus*), Caulimoviridae (genus *Badnavirus* and *Dioscovirus*), *Closteroviridae* (genus *Ampelovirus*), Potyviridae (genus *Macluravirus* and *Potyvirus*), and Secoviridae (genus *Sadwavirus*) have been reported to infect yams [2,18,19,20,21]. Typical symptoms of viral infection in yam may include stunting, mosaic, mottling, leaf distortion, necrosis, banding on leaves, and brown spots on the tubers (Figure 1) [22,23,24,25]. Whilst these viruses are predominantly spread through the cultivation of infected propagules, some can be transmitted through insect vectors, such as aphids and mealybugs, in a non/semi-persistent manner as well as through mechanical transmission [26,27,28].

The constraints of viral diseases on yam production necessitates urgent efforts for their identification, diagnosis, and management. Global efforts on yam virus characterization over the last five years have led to 13 novel virus species being recognized by the International Committee on Taxonomy of Viruses (ICTV) [19]. Whilst specific studies on yam viruses have focused on their identification, characterization, and diagnostic methods [12,13,24,29,30,31,32,33], broader studies have focused on virus incidence and distribution across different eco-geographical zones, and food and yield losses associated with yam virus infections [1,6,9,16,17,34,35]. The current gap in yam virus studies is the lack of an in-depth overview of key and prevalent yam infecting viruses in a single reference.

This review aims to provide an overview of yam infecting viruses by describing the most prevalent viruses reported to infect yam to date through discussions on their viral characteristics, disease symptoms, diagnostics, and sanitation methods. The information contained herein will provide insights for anyone working on developing diagnostic strategies and disease management in yam. It serves as a guiding document for researchers interested in conducting yam virus characterization, diagnostic and prevalence studies. There is an increased need to exchange yam germplasm to support breeding, evaluation for desired biotic and abiotic stresses, and increase production to support food and nutrition security. This review will serve as a point of reference for future global discussions on developing consolidated strategies for yam virus diagnostics to inform safe yam germplasm exchange. The existence of a consolidated general overview on yam viruses provides the opportunity to identify gaps in yam virus characterization and diagnostic protocols that need to be addressed to develop a health testing system to prevent the further geographical spread of yam viruses.

## 2. Family Potyviridae

The family *Potyviridae* is the largest family of RNA viruses to infect plants. There are eight genera in the family and members are distinguished by host range, genomic features, and phylogeny [36]. The virions are flexuous, filamentous particles ranging from 680 to 900 nm in length and 11 to 20 nm in width, whilst the genomes are single-stranded, positive-sense ribonucleic acids (RNAs) ranging from 8.3 to 11.3 kilobase (kb) in length. The typical genome of the family *Potyviridae* encodes a single large open reading frame (ORF), except for the genus *Bymovirus* which contains two ORFs. However, a second small ORF termed pretty interesting *Potyviridae* ORF (PIPO) was discovered embedded in an alternative reading frame [37]. The polyproteins are cleaved into nine or more functional proteins by the endonuclease. Viruses from two genera, *Potyvirus* and *Macluravirus,* are important viruses infecting yam and are discussed in the following.

### 2.1. Genus Potyvirus

The genus *Potyvirus* is the largest genus and the most extensively studied within the family *Potyviridae* [38]. To date, 176 species have been described and they contain some of the most economically important viruses, such as potato virus Y (PVY) and plum pox virus (PPV) [39]. Potyviruses are generally narrow in host range and are transmitted by over 200 species of aphids in a non-persistent and non-circulative manner [38,40]. The three main viruses infecting yam from this genus are described.

The yam mosaic virus (YMV) (species *Yam mosaic virus*) is the most prevalent and economically important yam virus, infecting both cultivated and wild yams [6,41]. It has been reported in all yam-growing regions of West Africa, the West Indies, and the Caribbean and is commonly found to infect *D. alata*, *D. rotundata,* and *D. cayenensis-rotundata* [41] but is currently absent from the Pacific region [42]. YMV was first reported to infect *D. cayenensis* from the Ivory Coast [26]. The complete genome sequence of the YMV Ivory Coast isolate (accession number: U42596) was reported by Aleman et al. [43] and comprises 9608 nucleotides (nt) in length. The complete genome of a second isolate, YMV-NG (accession number: MG711313) was determined to be 9594 nt in length and shared 85% sequence identity to the YMV Ivory Coast isolate [44]. The polyprotein is cleaved by endonuclease into ten mature and one fusion protein (P3-PIPO) critical for viral replication and movement (Figure 2) [37,40].

The virus is naturally transmitted through vegetative propagation of infected material but may also be transmitted through aphid vectors, such as *Aphis gossypii* (cotton aphids) and *A. craccivora* (cowpea aphids), in a non-persistent manner [26]. YMV could also be mechanically transmitted to other species, such as *Nicotiana benthamiana*, *N. megalosiphon,* and *Chenopodium amaranticolor* [2,41,45]. Symptoms on infected plants may include mottling, chlorosis of leaf and vein, interveinal mosaic, leaf distortion, and stunted growth [2,46]. Severe losses have been reported on YMV-infected *D. rotundata*, the most important food yam in West Africa [6]. The use of clean virus-free seeds and breeding for YMV-resistant planting materials remain the most effective method to mitigate the spread of YMV [25,33,47].

Understanding the genetic diversity of YMV isolates is critical for diagnostic tool development and yam improvement. The first analysis of diversity among YMV isolates identified six distinct serogroups based on symptomology, Western immunoblotting, and enzyme-linked immunosorbent assay (ELISA) [45,48]. Molecular evaluation of genetic diversity was assessed by sequencing the C-terminus part of the replicase (Nib), CP, and the 3′-untranslated region (3′UTR) of 27 YMV isolates of *D. alata*, *D. cayenensis-rotundata* complex, and *D. trifida* from Africa, the Caribbean, and French Guiana [49]. The CP region of YMV was the most variable compared to eight other potyviruses and phylogenetic analyses revealed nine distinct molecular groups, with the most diversified and divergent groups including isolates originating from Africa [41,50]. Azeteh et al. [1] reported the phylogenetic analysis of 27 YMV isolates from Cameroon based on CP sequences into three phylogenetic groups. However, these clustering did not correspond to agro-ecological zones or yam species and are likely attributed to the inter-zonal movement of planting materials and spread through aphid transmission.

Knowledge on the genetic mechanism of resistance to YMV in yam remains scarce and conventional breeding efforts for YMV resistance remain a challenge [23,51]. The inheritance of resistance to YMV was investigated in three tetraploid *D. rotundata* genotypes: TDr 93–1, TDr 93–2, and TDr 89/01444. Disease resistance and susceptibility in parental plants and progenies were scored on a visual disease severity scale of 1 to 5 where plants scored less than or equal to 2 as resistant and above 2 as susceptible as determined by the International Institute of Tropical Agriculture (IITA), Nigeria in 1998. Virus infection was further confirmed by TAS-ELISA. The F1 progeny segregation ratio indicated that resistance in TDr 89/01444 was governed by a single dominant gene in a simplex condition, whilst resistance in TDr 93–2 was associated with the presence of a major recessive gene in a duplex configuration. However, it was noted that the presence of mild mosaic symptoms and low titer of YMV in the resistant parental plants using TAS-ELISA and the observation of asymptomatic F1 progenies with high YMV titer may not represent true resistance but rather a form of tolerance in these genotypes [51]. Similarly, Bakayoko et al. [25] evaluated the resistance to YMV in 206 F1 progenies in the Ivory Coast based on the same disease severity scale reported by IITA (1998). It was reported that 91.3% of the progenies were considered resistant to YMV. However, this result was inconsistent with molecular analysis which revealed 41.7% of the F1 progenies with low severity scores were positive for YMV infection. It was suggested that 1-year-old F1 hybrid progenies were not reliable in breeding programs to evaluate viral resistance. A subsequent study by Mignouna et al. [46] identified two RAPD markers (OPW18_850_ and OPX15_850_) closely linked in the coupling phase with the *Ymv-1* gene and were mapped on the same linkage group. This represented the first DNA markers for YMV resistance in yam.

Several serological and nucleic acid-based diagnostic methods have been described for the detection of YMV, such as triple antibody sandwich enzyme-linked immunosorbent assay (TAS-ELISA), immunocapture reverse transcription-polymerase chain reaction (IC-RT-PCR), with RT-PCR as the most frequently used method for YMV detection, targeting the CP region of the YMV genome (Table 1) [50,52,53]. However, these methods are laborious and involve many successive steps for target detection [24]. A rapid YMV specific detection method by reverse-transcription recombinase polymerase amplification (RT-RPA) has been described by Silva et al. [24]. Results from RT-RPA were found to be reproducible and had a similar sensitivity to RT-PCR. However, RT-RPA has many advantages over RT-PCR, such as a rapid sample processing time of less than 30 min and a single incubation temperature (optimal 37 °C) for amplification. Similarly, Nkere et al. [33] reported a chromogenic detection method of YMV by closed-tube RT loop-mediated isothermal amplification (CT-RT-LAMP). YMV-positive samples were visualized by chromogenic detection with SYBR Green 1 dye, and the assay was reported to be 100 times more sensitive compared to the standard RT-PCR method.

The yam mild mosaic virus (YMMV) (species *Yam mild mosaic virus*) is the second most important potyvirus to infect yam after YMV [54]. It was first described as yam virus 1 (YV1) and Dioscorea alata virus (DAV) and is very prevalent in Africa, infecting *D. alata* [27,55]. YMMV is now classified as a distinct potyvirus infecting yams based on the ICTV criteria for potyvirus species demarcation [30,32,34]. The complete genome sequences of YMMV isolates range from 9521 to 9538 nt long, excluding the poly(A) tail. The Brazilian isolate encodes a polyprotein of 3084 aa with a large ORF that is cleaved into 11 functional proteins: P1, HC-Pro, P3, 6K1, CI, 6K2, NIa-VPg, NIa-Pro, Nib, PIPO, and CP [30]. The CP protein contains a DAG motif, located on the N-terminus typical of potyviruses and has an important function in aphid transmissibility [54]. YMMV differs from other potyviruses based on the ICTV criteria for species demarcation in the genus *Potyviridae* which stands at <76–77% nt and <80% aa identity in the CP region. The aa sequence identity in the CP region between YMMV and YMV is 57.1% [32].

Virus distribution is mainly through vegetative propagation of infected tubers or vines but transmission by the aphid vector *A. craccivora* has been reported [27]. The virus typically causes mild mottling and mosaic symptoms on *D. alata*, *D. cayenensis-rotundata* complex, and *D. trifida* but appears to be symptomless on *D. rotundata* [52]. Whilst YMMV does not pose significant constraints to yam production compared to YMV, its global presence necessitates an understanding of its dispersion and diversity. In the Pacific, YMMV has been reported from Fiji, Papua New Guinea, Solomon Islands, New Caledonia, and Vanuatu [32,34,42].

A phylogenetic analysis using the 798 nt 3′-terminal region of CP and 5′-terminal region of the 3′-non coding region of 36 YMMV isolates from Asian, Pacific, African, Amazonian and Caribbean origins placed them into eight distinct genetic clades that shared a common ancestor [34]. Another phylogenetic analysis of 18 isolates from Ghana and Nigeria placed these isolates into two major groups, however, this clustering did not correlate with the geographic origin of these isolates suggesting that there may have been an exchange of virus-infected tubers and seeds within the region [54].

Zou et al. [56] recently conducted a comprehensive study of YMMV genetic diversity on 86 isolates from West Africa, Asia, South Pacific, and America and clustered them into 14 distinct groups, indicating high genetic diversity among global YMMV isolates. Whilst YMMV isolates from China clustered into two distinct groups, isolates from other geographical regions were more diverse, particularly those of West Africa and Central America origin. The distinct geographical distribution of Chinese YMMV isolates suggested that germplasm exchange of Chinese yams with other groups was infrequent. In addition, phylogenetic analysis using whole-genome sequences of twelve YMMV isolates revealed four chimeric genome patterns, suggesting that recombination events are frequent among YMMV isolates [56]. This finding supports YMMV genome recombination first observed using 3′-terminal genome sequences by Bousalem et al. [34]. Nkere et al. [54] conducted phylogenetic analysis on 18 full-length CP sequences of YMMV isolates from Ghana and Nigeria and these were clustered into two major groups. Sequence comparison with reference YMMV sequences in the National Centre for Biotechnology Information (NCBI) GenBank clustered these 18 YMMV isolates into four monophyletic groups as per classification by Bousalem et al. [34].

Sensitive diagnostics that are able to detect the full spectrum of YMMV global diversity are imperative for safe germplasm exchange to prevent the exchange of YMMV isolates leading to unwanted recombination events that could lead to new virulent YMMV isolates [56]. Mumford and Seal [52] described the use of a rapid single-tube IC-RT-PCR for the detection of two yam potyviruses, YMV and YMMV, using species-specific primers (Table 1). The primers were highly specific to the target virus and did not show cross-amplification between YMV and YMMV. In addition, YMMV primers were able to amplify and detect a range of isolates, an important feature considering the high genetic diversity reported for YMMV. The test method also reported a thousand-fold increase in sensitivity compared to existing ELISA tests of the time [52]. Nkere et al. [54] assessed the YMMV status of 1530 samples from 140 locations in Nigeria and Ghana by RT-PCR using the YMMV-specific primer pair YMMV-CP-Bam and YMMV-CP-EcoRP (Table 1) and found 12.8% prevalence. In addition, Eni et al. [57] reported the use of a protein-A-sandwich ELISA using polyclonal antibodies for the detection of YMMV.

**Table 1 viruses-14-00662-t001:** List of yam viruses, their classification, and PCR primers available for their detection.

Virus Name	Genome Organization	Primer Name	Gene Region/ORF	Primer Sequence (5′–3′)	Annealing Temperature (°C)	PCR Product Length (bp)	Reference
YMV	(+) ssRNA	YMV1 YMV2	NIb, CP, 3′UTR	TGCGGAACTCRAAAGAAC TGCCATCAAATCCAAACA	52	196	[50]
		YMV1 Poly T	NIb, CP, 3′UTR	TGCGGAACTCRAAAGAAC TTTTTTTTTTTTTTTTTTTTTTTTV	52	450	[50]
		YMV CP 1F YMV UTR 1R	CP, 3′UTR	ATCCGGGATGTGGACAATGA TGGTCCTCCGCCACATCAAA	55	586	[52]
YMMV	(+) ssRNA	YMMV CP 2F YMMV UTR 1R	CP, 3′UTR	GGCACACATGCAAATGAAAGC CACCAGTAGAGTGAACATAG	55	249	[52]
		YMMVCP-Bam FP YMMV-CP-EcoRP	CP	CAGAGAGGATCCGCAAGTAAGGAACAGACATTTG TTGATCGAATTCCTAGATATTGCGCACTCCAAGAAG	55	798	[54]
Macluraviruses	(+) ssRNA	YamMac4F YamMac5R	CP	CHGCAGCWATYGGKMGTG GGGTTGCTGAGCRTYGGA	47	292	[19]
CYNMV	(+) ssRNA	CYNMV-Det-Fw CYNMV-Det-Rv	3′ proximal region	GTGTGCTAACAATGGTACATCATC GTGCGTTGAGGGTTGCTGAGC	55	609	[58]
YCNV	(+) ssRNA	YCNF504 YCNR1269	5′ region	TTAACGTCATAGTAGCTCTCAGC TAGTCGAGCTGAACCCTAACCAC	-	-	[12]
Badnaviruses	Circular dsDNA	BadnaFP BadnaRP	RT, RnaseH	ATGCCITTYGGIITIAARAAYGCICC CCAYTTRCAIACISCICCCCAICC	49	579	[59]
CMV	(+) ssRNA	CMV1-F CMV1-R	3′UTR, CP	GTAGACATCTGTGACGCGA GCGCGAAACAAGCTTCTTATC	55	540	[60]
		CMV primer 1 CMV primer 2	CP	GCCGTAAGCTGGATGGACAA TATGATAAGAAGCTTGTTTCGCG	55	500	[61]
		CMV-F IITA CMV-R	Replicase	GCCGTAAGCTGGATGGACAA CCGCTTGTGCGTTTAATGGCT	54	520	[1]
Potexviruses	(+) ssRNA	Potex-2RC Potex-5	ORF1	AGCATRGCNSCRTCYTG CAYCARCARGCMAARGAYGA	49	584	[62]
		YamX-3F yamX-5R	RdRp domain	CICARTGGGTIAAGAAARWKGA GRTCRAAIGCIGTRAARTCATTIGC	43	255	[19]
YaV1	(+) ssRNA	DiosClos-F DiosClos-R	3′end ORF8, 3′NCR	CTCTTTAGGTTTCCCATTTATCA TGGTTCTACATTACTAGACTAC	49	285	[63]
DMaV	(+) ssRNA	Seco-1F Seco-1R	RdRp	AACTCCWTCWGGWTTYGCTYTGAC CCCACTTYCTY TGAGAAAATCAA	46	323	[19]
YVY	(+) ssRNA	YVY-RdRp1-PF YVY-RdRp1-PR	RdRp	GTAATTGAAAATCACAGTGAGC CTTCAAGTGCATAATTGTCTAT	55	790	[20]
		YVY-CP-F YVY-CP-R	CP, 3′UTR	TTGATTAGTTAAGTATTTAGC CCAGTTTTTCCTGCTGGCAAAC	55	788	[20]
DNUaV	Circular dsDNA	DNUaV-ORF4-FP1 DNUaV-ORF4-RP1	ORF4 sequence	CCGGGTTGCCAGTACAGAAT CGTGAAGCACCCAAACCTTG	57	450	[18]

The Japanese yam mosaic virus (JYMV) (species *Japanese yam mosaic virus*) was first isolated from the Japanese yam, *D. japonica* in 1974 [31]. It was initially reported as a strain of YMV based on its potyvirus-like particle but was reclassified into a new potyvirus following genomic sequence characterization [31,64]. The complete genome of JYMV is 9757 nt long, excluding the poly(A) tail, with a single ORF encoding a polyprotein of 3130 aa. The genome organization of JYMV is typical of a potyvirus. Nucleotide sequence comparison of JYMV with YMV and YMMV showed 55.4% and 53.9% nucleotide identity, respectively. The identity is below the 76% species demarcation threshold of potyvirus, representing a new potyvirus species [64]. Current reports indicate that JYMV is restricted to China, Japan, and Korea and infects several members of the genus *Dioscorea* [31,64,65].

A novel strain of JYMV, designated JYMV-CN, was identified from the yam species, *D. polystachya,* from Yunnan Province, China [64]. The complete sequence has 9701 nt in length, excluding the poly(A) tail, encoding a polyprotein of 3247 aa. Nucleotide sequence comparison with two Japanese isolates revealed identities of 74.7–74.8% at the whole genome level, below the potyvirus species demarcation threshold of 76%. However, sequence analysis of the individual proteins suggested that the Chinese isolate is a divergent strain of JYMV and in the process of speciation [64]. A partial nucleotide sequence (7736 nt) of JYMV isolate BRI infecting *D. opposita* in Korea has also been recently published [65].

The virus is typically transmitted through vegetative propagation of seed tubers but may be transmitted by aphid vectors, *A. gossypii,* and *Myzus persicae*, in a non-persistent manner [31,66]. Symptoms of JYMV infection are similar to YMV, such as mosaic, green-banding of leaves, yellow stripes, and chlorosis, that can cause significant yield loss [64,67]

Fuji and Nakamae [31] reported the use of two methods, RT-PCR and double-antibody sandwich ELISA (DAS-ELISA) for the detection of JYMV. Comparison of the two detection methods showed that RT-PCR was more efficient in detecting JYMV than DAS-ELISA. In addition, DAS-ELISA was unable to detect all serotypes of JYMV. Similarly, Lee et al. [65] reported the use of an RT-PCR method for the diagnosis of a Korean JYMV isolate. Fukuta et al. [68] described a rapid and simple RT-LAMP detection method for JYMV based on the RNA extraction of Wang et al. [69] that allowed for the direct detection of RNA from infected plants without the need for pure RNA, precise thermal cycling, and gel electrophoresis.

A rapid and low-cost detection method for JYMV based on print-capture RT-PCR was reported by Mochizuki et al. [66]. Nucleic acid is captured onto a nitrocellulose membrane from leaf sap and recovered through an elution solution. Elution solution containing nucleic acid can then be directly used for RT-nested PCR for virus detection. This method offers many advantages in that it does not require toxic solvents and liquid nitrogen for nucleic acid extraction, the purified RNA can be used for RT-PCR operation, and tissue printed membranes can be stored for at least 3 months at 4 °C.

### 2.2. Genus Macluravirus

The genus *Macluravirus* resembles members of the genus *Potyvirus* in their transmission, but virions are slightly shorter (650–675 nm by 13–16 nm), lacking the P1 protein, a shorter HC-pro, and the absence of the DAG motif located in the CP for aphid transmission. The genus forms a distinct phylogenetic group within the family *Potyviridae* and has different consensus cleavage sites [12,70]. The virions contain one molecule of linear positive-sense single-stranded RNA (ssRNA) of about 8.0 kb. Three species from the genus, *Chinese yam necrotic mosaic virus*, *Yam chlorotic mosaic virus,* and *Yam chlorotic necrosis virus* have been identified infecting yam and are currently restricted to China, India, Japan, and Korea [10,12,71,72].

The Chinese yam necrotic mosaic virus (CYNMV) (species *Chinese yam necrotic mosaic virus*) was first reported as the causal agent of necrotic yam disease in the yam cultivar Naigamo (*D. opposita*) from Japan and was previously identified as a carlavirus based on its morphology and transmission by aphids [70]. However, partial genome sequencing of the 3′-terminal portions identified CYNMV as a distinct member of the genus *Macluravirus*. This was further supported by phylogenetic analysis of its CP amino acid sequences [10].

Complete nucleotide sequences of CYNMV isolates from Japan, China and Korea yielded genomic RNAs of ca. 8200 nt in length encoding for nine putative proteins that shared similar genomic organization to potyviruses (Figure 3) [10,58,73]. However, CYNMV lacks the P1 cistron counterpart and has a short HC-Pro cistron in the 5′ end compared to potyviruses, making macluravirus genomes the smallest in the family *Potyviridae* (Figure 3) [10]. Sequence alignment of CYNMV isolates also indicated the region containing the C-terminus of Nib and N-terminus of CP showed high variability. Kondo and Fujita [10] constructed the first full-length cDNA infectious clone of CYNMV under the control of the cauliflower mosaic virus 35S promoter and the nopaline synthase terminator that resulted in systemic necrotic mosaic symptoms of CYNMV in the yam variety *Nagaimo*. The infectious clones provide an opportunity to investigate infectivity, host range, symptom expression and virus localization with plant host.

Chinese yam necrotic mosaic virus can cause significant yield and quality losses, with yield losses of 30–45% being reported for CYNMV infection. It is easily transmitted by aphids in a non-persistent manner with a host range restricted to *Dioscorea* spp. [10]. Based on sequence comparison of CYNMV isolates reported, Kwon et al. [58] designed CYNMV-specific RT-PCR primer pairs, CYNMV-Det-FW and CYNMV-Det-Rv (Table 1), for the diagnosis of CYNMV.

The yam chlorotic necrotic mosaic virus (YCNMV), renamed to yam chlorotic mosaic virus (YCMV) (species *Yam chlorotic mosaic virus*) following ICTV convention, is the second yam-infecting macluravirus and was identified from the Chinese yam, *D. parviflora* [12,71]. It has a monopartite ssRNA of 8208 nt in length, excluding the poly(A) tail, and is currently the smallest genome reported from the family *Potyviridae* [71]. Its genome organization is typical of macluravirus, lacking the P1 protein, N-terminal HC-Pro, and DAG motif for aphid transmission found in potyviruses. Nucleotide sequence and polyprotein amino acid comparison showed that YCMV is most closely related to CYNMV and phylogenetic analysis grouped them into the same subgroup within the *Macluravirus* genus [12,71]. The biological characteristics of YCMV are still unknown.

The complete sequences of yam chlorotic necrosis virus (YCNV) (species *Yam chlorotic necrosis virus*), a third member of the yam-infecting macluravirus, were recently reported [12,72]. The first complete RNA genome reported by Lan et al. [12] consists of 8261 nt, with genome organization typical of the macluravirus. A second complete genome sequence of the YCNV isolate Kerala was reported by Filloux et al. [72] comprising 8263 nt in length. A basic local alignment search (BLAST) of the CP coding region in the GenBank database showed that the Chinese YCNV-YJish isolate had the highest similarity with an Indian yam-infecting macluravirus isolate YMCTCRI-03, with 85% amino acid sequence identity, indicating that these two isolates are representatives of the same species. Phylogenetic analysis using maximum-likelihood showed that YCNV is most closely related to CYNMV and YCMV [12]. Primer pairs YCNF504 and YCNR1269 developed by Lan et al. [12] (Table 1) were used to detect YCNV in 273 leaf samples of *D. alata* by RT-PCR. Seventy-two of the 273 samples tested positive for YCNV, indicating that YCNV is prevalent in Yunnan Province, China. In addition, YCNV can be mechanically transmitted to *Vigna unguiculata* (cowpea) and *Phaseolus vulgaris* (French bean) through sap inoculation [12].

Further macluravirus species may be present as identified by virus characterization work carried out on yam collections maintained by Guadeloupe’s Biological Resource Centre for Tropical Plants (CRB-PT) and additional samples from India and the South Pacific [74] The samples were screened for macluravirus using primers (Table 1) designed from the alignment of putative new macluravirus species identified from CAP3-assembled sequences of public ESTs (NCBI) and sequences of known macluraviruses. Sanger sequencing of RT-PCR amplicons revealed two novel macluravirus species in tropical yam, tentatively named Dioscorea alata macluravirus and Dioscorea esculenta macluravirus. These two species were present in samples from Nigeria, Guadeloupe, India, and some Pacific Islands (Palau, PNG, Tonga, and Vanuatu) [74].

## 3. Family Caulimoviridae

The family *Caulimoviridae* comprises non-enveloped reverse-transcribing plant viruses with non-covalently closed circular double-stranded DNA (dsDNA) genomes of 6.9–9.8 kb with two genera being encapsulated by viral CP into bacilliform-shaped virions, *Badnavirus* and *Tungrovirus*, whereas members of the genera *Caulimovirus*, *Cavemovirus*, *Petuvirus*, *Rosadnavirus*, *Solendovirus,* and *Soymovirus* have isometric virions. Yet, no virion morphology data are available for the genera *Dioscovirus* and *Vaccinivirus* [75]. Only members from two of the genera, *Badnavirus* and *Dioscovirus*, are known to infect yam.

The genus *Badnavirus* is a group of plant pararetroviruses that have been reported to infect a wide range of economically important plants, such as aroids, banana, black pepper, citrus, cocoa, gooseberry, grape, ornamental spiraea, red raspberry, sugarcane, sweet potato and yam of the tropical and temperate regions of Africa, Asia, Europe, Oceania and the Americas [76,77]. The Badnavirus genome comprises a single circular dsDNA ca. 6.9–9.2 kb encapsulated in a non-enveloped bacilliform particle of ca. 130 nm in length and 30 nm in width [9,78]. The genome contains at least three ORFs in the positive strand (Figure 4), with each strand having a single discontinuity, giving origin to three proteins P1, P2 and P3. The function of P1 remains to be elucidated, the P2 is the virion-associated capsid protein whilst the polyprotein P3 contains many functional domains, particularly, viral movement protein (VMP) (PF01107), coat protein (CP), retropepsin (pepsin-like aspartic protease) (AP) (CD00303), reverse transcriptase (RT) (CD01647) and RNase H1 (RH1) (CD06222) (Figure 4) [9,79,80].

Badnavirus-like particles were first reported from yam in association with a potyvirus causing internal brown spot disease in *D. alata* and *D. cayenensis-rotundata* complex in the Caribbean in the 1970s [81,82]. Subsequently, badnaviruses in other *Dioscorea* spp. were detected and partially characterized from several countries of West Africa, the South Pacific, and South America of which two were tentatively designated as species Dioscorea alata bacilliform virus (DaBV) and a serologically similar Dioscorea bulbifera bacilliform virus (DbBV) [17,28,83].

Badnaviruses are the most widespread viruses infecting yam and are collectively referred to as Dioscorea bacilliform virus (DBV) [17]. DBVs are vegetatively transmitted as well as by various mealybugs, *Planococcus citri*, with most DBV infections being symptomless, however, leaf symptoms, such as veinal chlorosis, necrosis, puckering, and crinkling, have been observed [17,28]. The first complete sequence of a DBV was obtained from a Nigerian *D. alata* and named Dioscorea alata bacilliform virus (DBALV) (species *Dioscorea bacilliform* AL *virus*) [28,84]. Later, the complete genome sequence of two additional isolates, representing a second species, of 7261 nt in length and sharing 61.9% sequence identity to DBALV was obtained from *D. sansibarensis* originating from Benin and named Dioscorea sansibarensis bacilliform virus (DBSNV)(species *Dioscorea bacilliform* SN *virus*) [85]. In the last decade, an additional six distinct genomes of DBV species have been completely sequenced, *Dioscorea bacilliform* AL *virus* 2 (DBALV2), *Dioscorea bacilliform* ES *virus* (DBESV), *Dioscorea bacilliform* RT *virus* 1 (DBRTV1), *Dioscorea bacilliform* RT *virus* 2 (DBRTV2), *Dioscorea bacilliform* RT *virus* 3 (DBRTV3) and *Dioscorea bacilliform* TR *virus* (DBTRV) [29,79,80,86].

Phylogenetic analysis of 80 partial reverse transcriptase (RT)-ribonuclease H (RNase H) coding sequences generated using the BadnaFP/RP [59] primers revealed 15 sequence groups with less than 79% nucleotide identity to each other and four divergent groups falling outside the badnavirus group but clustering outside ten currently recognized genera within the family *Caulimoviridae* [17,18,35,79,80,83,87,88,89]. Subsequent characterization of badnavirus infecting Pacific yam germplasm collections revealed the presence of DBALV, DBALV2, DBESV, and DBRTV2. The most prevalent virus in the collection, DBALV, was identified from samples originating from Vanuatu and Tonga, whilst DBALV2, DBESV, and DBRTV2 were found restricted to PNG, Fiji, and Samoa, respectively [77,86]. Further, integrated viral sequences, called endogenous pararetroviruses (EPRVs) have been identified from yams referred to as endogenous Dioscorea bacilliform viruses (eDBVs) [35,89]. The presence of badnavirus eDBVs in yam was first demonstrated in three *D. rotundata* samples from Guinea through hybridization studies [35]. The genomic organization of EPRVs can be complex, consisting of rearranged patterns of tandem repeats, fragmentations, inversions, and duplications of complete or partial viral genomes. [35,89]. Further, the genome of an African yam *D. cayenensis-rotundata* complex has been demonstrated to host eDBVs from four distinct badnavirus species (groups K5, K8, K9, and U12) [89]. However, it remains to be determined if they are transcriptionally active and potentially infectious [44].

Diagnosis of badnaviruses and development of tools for badnavirus detection in yam is complicated by the high badnavirus genetic variability [35]. The discovery of eDBVs into yam genomes further complicates the detection of yam badnavirus [89]. The presence of diverse eDBVs in yam genomes poses serious challenges in the differentiation of integrated sequences from the episomal virus with existing PCR-based diagnostic tools [80]. The high genetic diversity and lack of sufficiently specific and polyclonal antisera to DBV species is yet another constraint for immunocapture-PCR (IC-PCR) amplification and detection of episomal viruses [79,87].

Bömer et al. [79] reported the use of sequence-independent random primed (RP) rolling circle amplification (RCA) for the amplification of episomal (DBV) genomes in yam. This study resulted in the identification and characterization of nine complete genomic sequences of the existing and previously undescribed DBV phylogenetic groups from *D. alata* and *D. rotundata*. However, the study also highlighted the disadvantages of the sequence-independent nature of the RP-RCA. The use of random hexamers in the RP-RCA reactions also promotes the amplification of DNA of plant-origin, such as mitochondrial or chloroplast DNA [79,90,91], which precludes RP-RCA from being used as a routine diagnostic protocol. James et al. [92] and Sukal et al. [93] demonstrated that the inclusion of sequence-specific primers into the RP-RCA reactions creates a bias towards the target sequence. Further, Sukal et al. [93] showed the possibility of using specific-primed RCA coupled with restriction analysis as a potential diagnostic tool for DBV. The study further described RCA combined with next-generation sequencing (NGS) as a possible method for the diagnosis and characterization of badnaviruses. Turaki et al. [88] developed a PCR-dependent denaturing gradient gel electrophoresis (PCR-DGGE) workflow for the rapid and efficient determination and the unraveling of complex mixtures of potentially episomal and endogenous badnavirus sequences. This resulted in the identification of complex ‘fingerprints’ made up of multiple sequences of DBV. This technique can be particularly useful for badnavirus diversity studies.

The genus *Dioscovirus* is a newly described genus in the family *Caulimoviridae* and consists of a single species, *Dioscorea nummularia-associated virus* (DNUaV) [18]. It is a novel circular double-stranded DNA virus found infecting *D. nummularia* originating from Samoa. The genome of DNUaV was generated using rolling circle amplification (RCA), followed by cloning and sequencing of restriction endonucleases, *Eco*R1, *Kpn*1, *Sph*1, and *Stu*1, linearised products. The DNUaV genome is 8139 nt in length and contains four putative ORFs. Whilst ORFs 1 and 2 did not have identifiable conserved domains, ORF 3 had conserved domains, such as CP, MP, aspartic protease, and RT/RNase H1, typical of *Caulimoviridae* and ORF 4 contained a transactivator (TAV) domain (Figure 5) [18].

Primer pairs for PCR detection were designed by Sukal et al. [18] amplifying a 450 bp region of the putative ORF4 sequence and were used to screen a collection of 173 samples obtained from the Centre for Pacific Crops and Trees (CePaCT), a regional genebank based in Fiji. Among the 173 samples screened, only two Samoan *D. nummularia* samples generated the expected amplicons. The DNUaV positive plants did not show any disease symptoms and it is yet to be determined how it affects yam plants and associated yield.

## 4. Family Bromoviridae, Genus Cucumovirus

The family *Bromoviridae* contains some of the most important plant RNA viruses, with members distributed worldwide, infecting over 10,000 plant species [94]. The genomes of this family are tri-segmented, positive-sense, and single-stranded RNAs approximately 8 kb in length [95]. RNA1 and RNA2 encode for proteins involved in virus replication, whilst RNA3 encodes for the proteins, MP and CP (Figure 6). In some members, especially genus *Cucumovirus*, a fifth protein, P2b, located in RNA2 and part of the C-terminus region of the P2 protein, is involved in silencing suppression, systemic movement, and expression of symptoms (Figure 6) [94].

Cucumber mosaic virus (CMV) (species *Cucumber mosaic virus*) is a member of the genus *Cucumovirus*. It has an isometric single-stranded, positive-sense tripartite RNA genome consisting of RNA1, RNA2, and RNA3 and two subgenomic regions, RNA4 and RNA4A [21,96]. These are translated into five proteins, designated as 1a, 2a, 2b, 3a, and CP. In particular, the CMV 2b protein functions in long-distance virus movement, systemic symptoms expression and inhibition of virus silencing and is important for disease development [96]. It has a very wide host range, affecting more than 1200 plant species belonging to 100 families, and is transmitted by mechanical inoculation of plant sap and over 80 species of aphids in a non-persistent manner [21,97].

The first CMV infection in yam was reported as a virus disease in the yam *D. trifida* in Guadeloupe in the 1970s [98]. Currently, reports of yam CMV infections are restricted to West Africa infecting *D. alata* and *D. rotundata*. Field surveys conducted in yam growing regions of Benin, Ghana, and Togo reported their first record of CMV infections, albeit in low prevalence and mixed infection with other yam viruses [57,99]. On the contrary, a field survey of 591 leaf samples tested with multiplex RT-PCR revealed no positive CMV infection in yam in Cameroon [1]. Similarly, a germplasm collection of 38 *D. rotundata* maintained in the Ivory Coast tested negative for CMV despite previous reports of CMV infection in the Ivory Coast [25]. However, it was pointed out that the previous study was conducted on *D. alata*, whilst the current study was conducted on *D. rotundata*. Another prevalence study conducted on 396 accessions of yam from Guadeloupe for CMV also returned negative for CMV infection [19] despite previous reports of CMV infecting yam in Guadeloupe [98].

In the field, mixed infections of CMV with other viruses are common and symptoms are difficult to distinguish from single CMV infections [96]. Sap inoculation of yam CMV isolates induced systemic mosaic in *Cucumis sativus*, and systemic chlorosis, necrotic lesion, and leaf distortion in *Nicotiana glutinosa* [21]. Phylogenetic analysis of the 3′ nucleotide sequence of the CP gene and C-terminal noncoding region of RNA3 of a Benin CMV isolate categorized it as a subgroup 1A strain

ELISA is routinely used for the diagnosis of CMV in plants. Whilst antibodies are readily available commercially, serological differences between CMV isolates have been reported [60]. Eni et al. [100] reported the production of polyclonal antibodies against a yam isolate of CMV from Nigeria, which was able to detect CMV in infected yam leaves from Nigeria, Ghana, Togo, and Benin. Molecular detection using RT-PCR is the preferred method of detection and is generally based on generic CMV primers designed from other crops [1,60,61] (Table 1).

## 5. New Yam Viruses

In recent years, novel viruses from other plant virus families have been reported from yam. Viruses from the family *Alphaflexiviridae* (genus *Potexvirus*), *Betaflexiviridae*, *Secoviridae* (genus *Sadwavirus*), and *Closteroviridae* (genus *Ampelovirus*) have also been reported from yam [13,20,63,101].

Mambole et al. [13] reported the complete genome sequence of a novel potexvirus, yam virus X (YVX) (species *Yam virus X*), genus *Potexvirus*, family *Alphaflexiviridae,* isolated from *D. trifida* in Guadeloupe. The YVX genome is 6158 nt in length, excluding the poly(A) tail, and encodes five ORFs. A large ORF1 encodes the RdRp, whilst ORFs 2, 3, and 4 encode the putative triple gene block proteins, TGBp1, TGBp2, and TGBp3, typical of potexvirus and function in viral movement. ORF5 encodes the CP. A BLAST search of the RdRp and CP amino acid sequences in the NCBI database and phylogenetic analysis confirmed this as a potexvirus but the maximum sequence identity was only 51.9% indicating it was a new member of the genus. Yam plants infected with YVX were symptomless except for a single plant co-infected with a potyvirus that showed mild symptoms. Mechanical transmission of viral isolates to indicator plants *N. benthamiana*, *N. clevelandii*, *C. quinoa,* and *C. amaranticolor* did not yield local or systemic infection and RT-PCR detection with the potexvirus-specific primers also yielded negative results. A prevalence study carried out by Mambole et al. [13] on 383 yam accessions from 34 countries reported 17 YVX positive samples using the potexvirus-specific degenerate primer pair Potex-2RC/Potex-5 (Table 1). Phylogenetic analysis of sequenced amplification products provided evidence for two additional and distinct groups of potexvirus sequences; group one includes sequences from *D. nummularia* from Vanuatu and group two includes sequences from *D. bulbifera* and *D. rotundata* from Haiti, and *D. trifida* and *D. rotundata* sequences from Guadeloupe.

The complete genome sequence of a novel putative secovirus was isolated from yam plants exhibiting mosaic symptoms in Brazil [101]. The genome is composed of two positive-sense RNA molecules of 5979 and 3809 nt in lengths, each with a single large ORF. RNA1-ORF1 was predicted to encode a polyprotein associated with the replication process. RNA2-ORF2 was predicted to encode for the CP and MP. It is tentatively called Dioscorea mosaic-associated virus (DmaV) of the genus *Sadwavirus*, family *Secoviridae*. Phylogenetic analysis of the protease–polymerase (Pro-Pol) amino acid sequence of DmaV with other members of secovirus indicated that it most closely related to the chocolate lily virus A (CLVA), whilst the amino acid sequence identity indicated it was a putative new member of the family *Secoviridae* based on ICTV species demarcation criteria.

Marais et al. [63] recently reported the complete nucleotide sequence of a novel yam virus tentatively named yam asymptomatic virus 1 (YaV1) (species *Yam asymptomatic virus 1*) from an asymptomatic *D. alata* plant collected from Vanuatu. The genome is 14,855 nt in length, encoding for 10 putative ORFs with a similar organization to that of the little cherry virus 2 (LChV2) of subgroup 1 of genus *Ampelovirus.* Phylogenetic analysis of the HSP70 and CP amino acid sequences confirmed it to be a novel member of the genus *Ampelovirus*, family *Closteroviridae* and distinct from another recently identified ampelovirus, air potato virus 1 (AiPoV1) infecting *D. bulbifera* from Florida, USA [102]. DiosClos-F and DiosClos-R primer pair were used to screen a yam field collection of 170 accessions in Guadeloupe (French West Indies) by the Biological Resource Center for Tropical Plants (BRC-TP). A total of 86 accessions representing different yam species were positive for YaV1 and showed that YaV1 was highly prevalent in the field collection, but asymptomatic. The infected accessions were mostly of Caribbean origin with two accessions from Africa (Ivory Coast and Nigeria) and one from the Pacific (New Caledonia), suggesting that YaV1 may also be present in these countries. Future prevalence studies of YaV1 will be required to determine their geographical distribution [63]. Blastn analysis of the YaV1 genome sequence yielded two expressed sequence tags (ESTs) from a Nigerian *D. alata* plant, confirming its presence in Africa. Sanger sequencing of fifty-five selected amplification products was used for phylogenetic analysis but revealed low variability (93.3 to 100% sequence identity) amongst distinct accessions, suggesting that plant-to-plant transmission through insect vector may be possible.

The complete genome sequences of two isolates of the tentatively named ‘yam virus Y’ (YVY), obtained from a collection of *D. rotundata* from the Natural Resources Institute (NRI, UK), IITA, and Council for Scientific and Industrial Research—Crop Research Institute (CSIR-CRI, Ghana), were sequenced using high-throughput sequencing (HTS) [20]. The genomes of YVY-Dan and YVY-Mak isolates are 7557 nt and 7584 nt in length, respectively, excluding the poly(A) tail. The genome encodes five ORFs; ORF1 encodes a large replication protein, ORF2, ORF3 and ORF4 constitute the triple gene block protein, associated with viral movement, whilst ORF5 encodes a putative CP protein. Based on ICTV demarcation criteria and phylogenetic analysis, YVY is grouped with unassigned members of the family *Betaflexiviridae* and most closely related to sugarcane striate mosaic-associated virus (SCSMaV) (*Sugarcane striate mosaic-associated virus*). A prevalence study using newly developed YVY-specific PCR primers reported 31 YVY positive samples in a collection of 55 breeding lines and landraces grown in NRI UK, IITA Nigeria, and CSIR-CRI Ghana. Among these, 23 showed mixed infection with YMV. Plants that were singly infected with YVY were generally symptomless except for one plant, whilst plants infected with YMV or mix-infection developed symptoms [20].

## 6. Yam Virus Sanitation

Plant viruses are obligate intracellular parasites that can survive only inside living cells [103]. Once the viral infection is established in plants, it is not feasible to cure, in contrast to bacterial or fungal infections where they can be treated with antibacterial or antifungal agents [104]. Virus elimination through in vitro culture techniques has been proven to be successful in producing virus-free plants. Some of these established methods include shoot–tip or meristem culture, micrografting, chemotherapy, thermotherapy, and shoot–tip cryotherapy [103,105,106].

Shin et al. [107] reported the production of YMV-free *D. opposita* plants by cryotherapy of shoot tips. Shoot apices were precultured for 16 h in 0.3 M sucrose, encapsulated in sodium alginate, and dehydrated for 4 h before direct immersion in liquid nitrogen. Regenerated shoot tips were reported to be 90% YMV-free. Similarly, Ita et al. [2] reported the elimination of YMV from *D. rotundata* by cryotherapy of axillary buds of infected stocks. Enlarged axillary buds of infected plants were frozen in liquid nitrogen for one hour, re-warmed at 40 °C, and cultured to regenerate plants. Plantlet regeneration was reported at 76%, whilst YMV elimination was reported at 100%. Similar protocols can be adopted for other yam viruses

Hot water treatment was successfully used for the elimination of YMMV in *D. alata*. Single node vine cuttings were treated at 32 and 36 °C for different time durations and virus elimination was confirmed by RT-PCR using YMMV-specific primers. Treatment at 36 °C for 30 min was reported to be most efficient at 90% elimination [108]. Umber et al. [19] reported the use of a combination of thermotherapy and meristem culture for the elimination of yam viruses prevalent in Guadeloupe (badnavirus, YMV, YMMV, CMV, DMaV, YaV1, potexvirus, and macluravirus). Sanitation rates were variable among the different viruses, with YaV1 reporting the lowest at 14.5% and macluravirus the highest at 100% elimination. Among the 57 accessions subjected to combined thermotherapy and meristem culture protocol in the study, sixteen accessions were fully sanitized.

The application of water-dissolved ozone was reported for the sanitation of potyvirus during in vitro propagation of *D. cayenensis-rotundata* [109]. Potyvirus-positive nodal segments were subjected to different concentrations of water-dissolved ozone under different time durations. Treatment with 1.5 ppm ozone for 10 min was most efficient and produced 63% potyvirus-free in vitro yam plants. In addition, it was reported that this treatment stimulated plant tissue growth, thus reducing the time for the establishment stage during in vitro culture [109].

## 7. Yam Virus Status in the Pacific

Efficient virus diagnostic tools and sanitation are essential to facilitating germplasm exchange. The Centre for Pacific Crops and Trees (CePaCT) of the Land Resource Division (LRD) of the Pacific Community (SPC) located in Fiji is the premier genebank of the Pacific supporting the safe conservation and distribution of plant genetic resources of importance to the region. The center (CePaCT) has a unique yam collection from the Pacific region that is invaluable towards mitigating biotic and abiotic stresses in production. However, the yam diversity remains unavailable due to the fact that diagnostic protocols for yam viruses, particularly those endemic to the Pacific region, remain undefined.

In the 2000s, virus characterization work carried out in the region, though sporadic, has speculated the existence of known and novel badnavirus and potyvirus diversity in the region [17,32,34,42,110]. This has led to further characterization work on the diversity of badnaviruses, with the identification of two novel and two known badnaviruses [77,86]. The previous studies also highlight the need to carry out more characterization work in the region to delineate the diversity of viruses infecting Pacific yam collections and to develop diagnostic protocols to enable the testing of the germplasm before exchange. The diagnostic and sanitation protocols described in this review will greatly assist in the development of diagnostic and prevalence studies of yam viruses in the Pacific region to better understand their distribution and to facilitate the safe exchange of Pacific yam germplasm.

## 8. Conclusions and Perspectives

In this review, the most prevalent viruses infecting yam globally have been described for their origin, morphology, genome organization, symptoms caused, and diagnostics. As a predominantly vegetatively propagated crop, the use of clean, virus-free planting materials is the most effective method to curtail the spread of yam viruses. A major drawback, however, is the lack of formal seed systems within smallholder farming communities where globally, most yams are cultivated. The use and sharing of virus-infected planting materials by farmers promote the spread of yam viruses to new ecogeographical locations.

Yam viruses, such as YMV, YMMV, and badnaviruses, have been reported to be highly prevalent and genetically diverse. In addition, the discovery of integrated viral sequences from badnaviruses within host genomes adds a level of complexity to their diagnostics. As more yam-infecting viruses are being discovered from other virus families, the development of sensitive and specific diagnostic tools for their detection becomes paramount. This leaves the conundrum and predicament of whether a truly ‘virus-free’ plant can be achieved and what level of sanitation is required to meet international standards for germplasm exchange.

Serological and PCR-based methods of detection will continue to be the backbone of yam virus diagnostics. Their limitations, however, are that they can only detect known or related viruses. In addition, the high genetic diversity of virus isolates can limit their detection spectrum. The use of HTS technologies for sequencing and discovery of existing and novel yam viruses is gaining traction. Bomer et al. [44] evaluated a combined tissue culture and NGS approach for the detection of yam virus without prior knowledge of viral sequences and successfully detected and sequenced two novel badnaviruses and one novel YMV isolate. However, the practicality of HTS for routine virus diagnostics is yet to be realized. The use of a Nanopore-based MinION approach demonstrated its ability to both reliably detect and sequence near full-length genomes of yam viruses [72], representing an important first step for future research into portable field-based diagnostics

Virus elimination through in vitro culture techniques has proven effective for producing virus-free plants in yam. Natural resistance, however, might prove to be more economic and efficient in the long term. Work on genetic resistance and crop improvement has been slow in the past due to a lack of genetic and genomic tools. With the recent publication of several whole-genome sequences of *Dioscorea* spp. [3,111,112], and the discovery of thousands of good quality single nucleotide polymorphisms (SNPs) [113], research in areas such as gene mapping, marker development for marker-assisted breeding, virus-host interaction, and molecular mechanisms of resistance in yam should start gaining momentum.

This is the first review of global yam viruses and the information contained herein will invaluably facilitate further development in yam virus diagnostics and sanitation for the safe international exchange of yams particularly in the Pacific and those held elsewhere in the world.

## Figures and Tables

**Figure 1 viruses-14-00662-f001:**
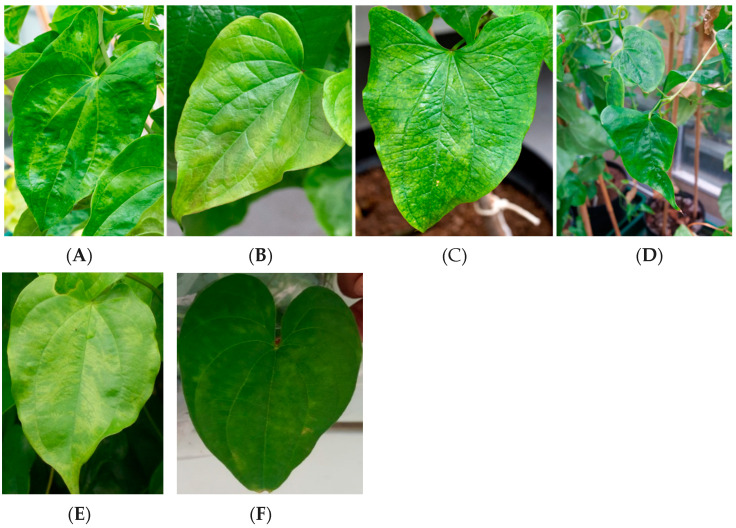
Typical virus symptoms on yam. (**A**) *D. rotundata* TDr 99/02674 showing mosaic symptoms; (**B**) *D. rotundata* cv. Adaka showing chlorotic leaf discoloration, and (**C**) mottling; (**D**) *D. rotundata* TDr 00/00168 showing leaf deformation ((**A**–**D**) source G.R.E. Silva (NRI) [20]); (**E**) *D. rotundata* leaf discoloration (Source: Stephen Winter, DSMZ); (**F**) *D. alata* showing diffused mottling.

**Figure 2 viruses-14-00662-f002:**
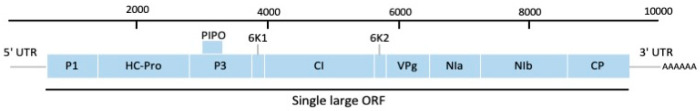
Schematic genome representation of species *Yam mosaic virus* (YMV), genus *Potyvirus*, family *Potyviridae*. The 11 functional proteins represented are P1: first protein (translation, modulator of replication), HC-Pro: helper component protease (silencing suppression and aphid transmission), P3: third protein (virus replication and movement), 6K1 (formation of replication vesicles), CI: cylindrical inclusion protein (virus movement and replication), 6K2 (formation of replication vesicles), VPg: viral genome-linked protein (translation, virus movement and replication), NIa: major protease of small nuclear inclusion protein (polyprotein processing), NIb: large nuclear inclusion protein (RNA-dependent RNA polymerase), CP: coat protein (virus movement, virion formation, and aphid transmission) and PIPO: pretty-interesting-potyvirus-ORF (virus movement).

**Figure 3 viruses-14-00662-f003:**
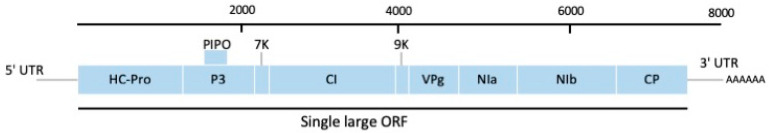
Schematic representation of species *Chinese yam necrotic mosaic virus* (CYNMV), genus *Macluravirus* genome, family *Potyviridae*. The 10 functional proteins represented are HC-Pro: helper component protease (gene silencing and vector transmission), P3: third protein (virus replication), 7K, CI: cylindrical inclusion protein (virus movement and replication), 9K, VPg: viral genome-linked protein (viral replication), NIa: major protease of small nuclear inclusion protein (polyprotein processing), NIb: large nuclear inclusion protein (RNA-dependent RNA polymerase), CP: coat protein (virus movement and virion formation) and PIPO: pretty-interesting-potyvirus-ORF (virus movement).

**Figure 4 viruses-14-00662-f004:**
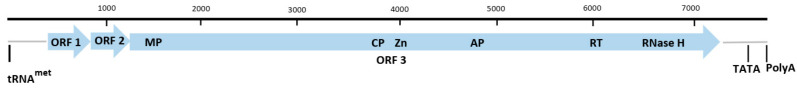
Schematic representation for the linearised genome organization of members of the genus *Badnavirus*, family *Caulimoviridae.* ORF1 encodes P1 protein (function unknown), ORF2 encodes P2 virion-associated capsid protein (nucleic acid binding activity), ORF3 encodes polyprotein P3 (VMP, CP, AP, RT, RNase H1). tRNA^met^ binding site designated as the origin of the circular genome.

**Figure 5 viruses-14-00662-f005:**
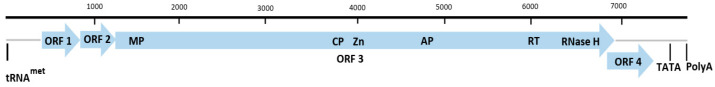
Schematic representation for the linearised genome organization of species *Dioscorea nummularia-associated virus* (DNUaV) genus *Dioscovirus*, family *Caulimoviridae*. ORF1 encodes P1 protein (function unknown), ORF2 encodes P2 (function unknown), ORF3 encodes polyprotein P3 (VMP, CP, AP, RT, RNase H1) and ORF4 encodes P4 (TAV). tRNA^met^ binding site designated as the origin of the circular genome.

**Figure 6 viruses-14-00662-f006:**
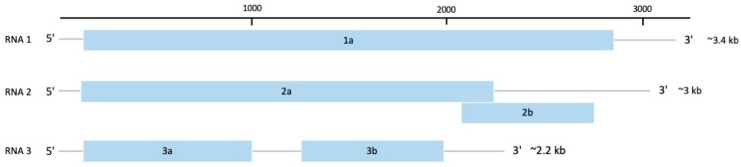
Schematic representation of the genome of species *Cucumber mosaic virus* (CMV), genus *Cucumovirus*, family *Bromoviridae*. The proteins encoded by each RNA are 1a: replication protein, 2a: RNA-dependent RNA polymerase, 2b: silencing suppressor, 3a: movement protein, and 3b: capsid protein.

## Data Availability

Not Applicable.

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
