# Peer review of "A Review of Viruses Infecting Yam (Dioscorea spp.)"

_viruses, 2022, doi:10.3390/v14040662_

Round 1
Reviewer 1 Report
The manuscript Viruses-1488389 describes all the viruses that infect yam and explains how this diversity affects the diffusion of yam resources worldwide.
If this review is well structured, the content is clearly not balanced, with a very big part on Potyviridae, a big part on Caulimoviridae and the rest is hardly overviewed. This is the biggest problem of this article, which lists a number of facts without really looking deep into them. There is no analysis, no synthesis and no illustrations that would give a global view of the implications of viral diseases on yam cultivation. In addition, the citations are not always appropriate, and be careful to cite the source article of an information, and not an article that cites the source article, which does not make sense. Finally, the description of viral species refers to all viruses that infect yams worldwide, not just in the Pacific, so why would viral diseases only affect the exchange of genetic material in the Pacific? I think it is easily possible to reframe the implications of these viruses at the global level. In conclusion, this manuscript should be more striking to be truly a review on yam viruses.
Please, see below some specific and general comments and some suggestions:
- Introduction:
- 31-34: The referenced articles are only about Africa, find references for other parts of the world.
- 36-38: The real source of this information is FAOSTATS. Do not cite articles that cite the source…
- 44-48: Betaflexiviridae: why name Carlavirus if in the section about macluraviruses you say that CYNMV is in fact a macluravirus? You should rather talk about YVY. Caulimoviridae: you miss Dioscovirus. And in the references, CMV, DNUaV and YVY are missing.
- 48-50: Add the reference 32 that describes some symptoms in Africa.
- 50-52: For the references, you have to find the source articles that prove the vector transmission.
- 62-65: The purpose of this review is not so clear. I did not find the “future needs for safe exchange” in the manuscript.
- Family Potyviridae:
Genus Potyvirus:
YMV:
- l.89-94: I don't see the interest of redoing the genomic characterization of all the viral species described in this paper. They have already been done in the papers that describe the species for the first time. Also, I do not understand what the figures of the genomic organization of these viruses bring to the review.
- 101-103: The reference for the vector transmission is missing.
- 107-108: The citation 31 is not the source of impact studies. But reference 6 yes.
- 111-120: Are there no more recent diversity studies for YMV?
- 121-122: Remove the reference 2, this is not the source. Are you sure that YMV resistant hybrids have been created? Isn't it more a matter of tolerance? Look at reference 32 on this subject, which should also be cited in this section about breeding.
- 130&133: The references are not good; it seems that there is a shift: 29=30 and 19=20. Be careful with this kind of errors before submitting an article.
- 136-139: You said that IC-RT-PCR is the detection tests more frequently used for YMV detection, but you cite an article from 20 years ago! How is this virus being tested now? What about RT-PCR, and the sets of primer used for YMV detection. Moreover, these primers are described in Table 1 whereas this table is not cited in this part on YMV,
- 146: The shift… 30=31.
YMMV:
- 151-156: What is DAV? If it is a potyvirus species, why there is no paragraph about it? It seems to me that DAV has been renamed YMMV instead, so it needs to be better explained and to give the proper references.
- 167&169: Indirect citations (31 and 44). Please give the source articles.
- 181-184: Please provide a reference. If it is the reference 44, place it after the first sentence and not the second, as is the case.
- 181-190: There is no need to rewrite the conclusions of Zou et al. [44] and indeed, if you analyze their phylogenetic tree carefully, you will see that phylogenetic groups are not really associated with geographic distribution, but that indeed YMMV has a strong propensity to recombine. So these sentences need to be rewritten.
- 191-192: Note that the YMV primers designed by Munford and Seal [45] do not detect all isolates of this virus, according to Bousalem et al. [36].
- 197-199: Why talk about prevalence for YMMV on a particular study when there was no such analysis on YMV?
- 200-203: The sentence must be in the YMMV diversity section (l.176-190).
Table 1:
- Potexviruses, YamX-1F/YamX-5R: only YamX-3F/YamX-5R are used for potexviruses detection. Please read Umber et al. [18] again.
JYMV:
- 217-218: Please provide a reference.
- 227-229: “that may cause” What does it mean?
- 230-237: What about the diagnosis method from Mochizuki et al. [56]?
- There are no diversity studies for JYMV?
Genus Macluravirus:
- There is a primer pair listed in Table 1 seems to detect yam macluraviruses. Yet there is no reference to them in the section on macluraviruses. What virus do they detect?
- 247-256: put the reference [61] at the end of the first sentence.
CYNMV:
- 257: mistake in the title: CYNMV and not JYMV.
- 263: remove reference 58.
- 284-287: What diagnostic method is used to detect this virus?
YCMV:
- There seems to be little data on this virus. As a result, the size of the paragraphs is very variable from one virus to another and this makes the paper unbalanced. Moreover, this virus has been renamed YCMV rather than YCNMV, according to the ICTV classification [9].
- 288: You forgot the virus name abbreviation in the title.
YCNV:
- 298: You forgot the virus name abbreviation in the title.
- 301-304: What is the demarcation line of the macluravirus genus? Whit 85% of AA identity in CP, it seems that YCNV is of the same species as this YMCTCRI-03 isolate, from which we do not know where it comes from. Please rewrite and explain.
- Family Caulimoviridae:
- 317: [63] is still an indirect citation. The official ICTV reference for the family Caulimoviridae is Teycheney et al. https://dx.doi.org/10.1099%2Fjgv.0.001497.
Genus Badnavirus:
- Why did not you separate Badnavirus into several species, as with Potyvirus and Macluravirus? Explain.
- Compare to Potyvirus section, Badnavirus section seems reduce in some points, like prevalence, global geographic distribution, symptomatology, vector transmission, while data exists on these topics. This section could be more detailed.
- 355-363: It is necessary to develop this paragraph, explain that each phylogenetic group corresponds to a species and that some groups represent episomal sequences and others not. Concerning the groups that are not attached to the genus Badnavirus, read Diop et al. https://doi.org/10.1038/s41598-017-16399-x.
- 371-373: Another indirect quote. Find the source articles that prove EPRV activation in banana.
- 373-374: This is reference 80, which shows that yam EPRVs belong to 4 distinct species, it is in the title…
- 374-376: Choose reference 68 rather than 80, where the IC-PCR test to detect Badnavirus was developed.
- 394-409: I do not understand why the NGS technique is described in this part of the article when it does not only detect badnaviruses, as indicated, since Bömer et al. found YMV and Filloux et al. YCNV. Furthermore, I do not think that it is reasonably possible to use NGS techniques as a large-scale detection tool. But perhaps a discussion around this point could be developed in section 6.
Genus Dioscovirus:
- Explain why you have no data on diversity, prevalence, distribution, symptomatology and transmission. Specific primers detecting DNUaV exit, so why did not you add them to the list of detection primers in Table 1?
- 419-420: The virus name abbreviation is missing a "U"…
- Family Bromoviridae:
Genus Cucumovirus:
CMV:
- You cannot have a section 4.1 without a section 4.2.
- 441-444: Indirect citations.
- 445-446: The reference is an article in 2010 about production of antibody in Africa! Please cite the source article from Guadeloupe in 70s (in French): Migliori, A. (1977) Maladie à virus de l’igname (Dioscorea sp.). In Proceedings of the 14th Annual Meeting of the Caribbean Food Crops Society, Guadeloupe and Martinique, France, 27 June-2 July 1977; pp. 428-435.
- 451-455: Add the citation after the first sentence of this part. A prevalence study in Guadeloupe showed the same result [18].
- 465-466: You cannot use the future for a result from an article that is 10 years old!
- 466-468: Explain that all these primer sets was not design from yam isolates of CMV, but from others plants. References should be numbered.
- Emerging yam viruses:
- This section could be greatly improved by going deeper. Finding many new viruses on yam by the NGS technique allows knowing better the virome of this plant, but also leads to many questions. What is their diversity, their prevalence, their geographical distribution, their impact? Do we have this kind of data on these 4 new viruses? The answer is yes, for some of them, so it is necessary to talk about it, to develop this section, which should be essential.
- 474-476: No need to put the same citation twice in the same sentence!
- 481-485: And the putative new potexvirus species from Côte d’Ivoire [32]?
- 490-492: The reference 51 talks about YaV1. I think it is a mistake.
- Yam virus diagnostics and elimination:
- As many diagnostic methods are common through yam viruses (except primers, of course), why did not you describe these methods only in this section instead of in the family sections, as it is repetitive.
- All of the sanitation methods described in this section focus on the elimination of Potyvirus (YMV or YMMV), while Umber et al. appear to have undertaken a broader evaluation of their sanitation method. On which viruses did they test this method, besides YaV1 and macluraviruses?
- 558: macluraviruses. The primer pair can detect several species.
- Conclusion and perspectives:
- 568: “yam viruses of economic importance have been described” On what basis can you say that the viruses described are of economic importance? There are yield data for a very small number of them and some even appear to be asymptomatic, like YaV1!
- 576-577: Integrated viral sequences, not genomes. Endogenous viral sequences sometimes contain several viral genomes, sometimes only a few parts of this genome, with most of the time very important rearrangements.
- 580-582: This is the crucial issue!
- 584-585: The reliability of these tools also depends on the detection spectrum of the antibodies or primers that may have difficulty targeting certain serotypes or variants.
- 590: “field-based plant virus diagnostics” Do you have any reference on the use of minION technique directly in the field? This technique is still complex to set up...
- 600-604: Do not forget genotyping of D. alata with SNP from Cormier et al. https://doi.org/10.1002/ece3.5141

Reviewer 2 Report
This review provides information about the different viruses being reported in yam (Dioscorea spp.) to date, giving more detailed information to the most important group of viruses: Potyvirus and Badnavirus.
I’d like to thank the authors for this review as it puts the relevant information regarding viruses in yam in a single place and can be very useful for anyone working with these viruses.
However, in my view, this review is just a statement of facts, where all the information is described in a light way. It’s not clear what is the main message of the review. Title suggest that this review will explore the implications of viruses in yam to the Pacific region but this is discuss very briefly at the end of the manuscript.
Authors did not explore research gaps neither discuss the importance or the challenges imposed by the different viruses. An example of a question to explore, amongst several others, would be the impact of multiple infections for seed multiplication centres. But again, this is related to the main objective of the review and that message is not clear. I strongly suggest authors to make this information clearer.
More specific comments related to the text:
line 78: symptom is “mosaic” and not “mosaicking”. please change throughout.
line 78: “Typical symptoms of virus in yam”: why not include pictures?
line 53: what is the economic impact of viral diseases on yam production?
line 54: “a number of novel viruses” – please indicate how many
lines 57-61: references are needed! What studies exist on host-virus interaction on yam?
line 79: give example of the most economically important potyvirus
section on YMV:
- authors only mentioned 1 complete genome. A quick NBCI analysis showed at least one more complete YMV genome. how do these genomes compare?
- what about the PIPO protein? this is not mentioned in the text (line94).
- figure 1: there are a lot of schematic representation of potyvirus genomes. This figure does not bring novelty to the review. If authors want to keep it please describe what is the function of the different proteins. This comment can be applied to all figures of the manuscript!
- regarding resistance to YMV: authors mention that “YMV resistant planting material remains the most effective method to combat YMV spread”. Are there any YMV resistant variety available to farmers? If yes please include references. Also, Odu et al considered resistance of a plant not showing symptoms of disease at the time of sampling but still YMV positive. In my view this is not resistance! Please redo the information about resistance to YMV.
- line 139: RT-PCR is the most common method to detect YMV and is not described here. please rephrase it.
section on YMMV:
- where is the DAG motif located? Please add info in the text (line 164)
- line 172: “…YMMV is subjected to a rapid evolution…” – there is no data regarding this. with no references to support this statement, this is speculation and should be removed or rephrased.
- line 180 and line 184: two studies with contradictory information. why? please add information.
- line 181: “a recent comprehensive study on YMMV” – which study? please include reference
- there are more types of diagnostics for YMMV. Please add those to the text
section JYMV:
- please add more information. how identical is JYMV to YMV and YMMV? why are these viruses different species?
- what is the distribution of the virus?
- line 237: please be more specific on what type of extraction is done
Macluravirus:
- no need to separate different viruses in sub sections. Merge text to improve flow.
- line 240: “… virions are slightly shorter”. please be more specific!
- lines 245-246: not true! YCNV was reported by Filloux et al 2018 on a yam plant from India. please correct this sentence.
- line 257: why “Chinese” in the name if virus was identified in Japan in a Japanese cultivar? Please clarify.
- lines 270-272: what is the point of this sentence?
- lines 272-275: why is this important?
- no other types of diagnostics for CYNMV?
- line 300: Filloux et al 2018 also reported complete sequence of YCNV. please add this information.
- line 306: what study?
Badnavirus:
- line 327: some badnavirus can have more that 3 ORFs… please rephase this.
- lines 364-379: this paragraph is confusing. jumps from yam to banana…! please rephrase it.
- line 399: why is a potyvirus being mentioned here? this info is out of place and text needs re-arrangement.
Dioscovius:
A lot of information missing! Detected where? Obtained by HTS? Any symptoms associated with the virus? what is the importance and what are the available diagnostic methods?
section 6
what is being done in the Pacific region? how are these elimination methods being employed?
section 7
- line 585: change NGS to HTS
- line 591: multiplex PCR is not new! there are a lot of published methods. for example Gambino and Gribaudo 2006 developed a multiplex PCR that detects 9 grapevine viruses. Please rephrase the information in the manuscript.
line 601: there are more yam genomes available… please give up do date information
As I mentioned above, the information provided in the manuscript is very light and I strongly suggest authors to state at the beginning what is the main objective for this review. For me is a statement of facts (some of them need to be revisited) and does not correlate with information described on lines 62-65.
I recommend the review for publication after addressing my comments above.
Round 2
Reviewer 2 Report
Thank you for providing a revised version of the manuscript. The message on the revised version is more clear and the review clearly mention that the point is to have a general overview of the current knowledge on the most important virus in yam in a single document.
This is an improved version of the manuscript, with each section written in a more clear and concise way, and table and figures contribute to a "better looking" manuscript.
I have however a few extra comments that I'd recommend Authors to address before publication.
Introduction
line 36: yam is a tuber crop. not a root crop. please add "root and tuber crop"
lines 75-81: this sentence needs revision. is too long and difficult to read. please redo sequence.
line 83-84: please add one sentence on how this review will address that.
section 2
please remove "family potyviridae" from title on section 2.1 and 2.2
please check virus names throughout. Capital 1st letter and if referring to a species virus name should be in italic
line 137: replace "distributed" by "transmitted by"
line 164: replace "susceptivity" by "susceptibility"
line 211: please add what is the ICTV species demarcation criteria and the identity % between YMV and YMMV
lines 237-240/lines 255-259: could this be merged? it seems information is jumping forward and backwards and I think flow could be improved
line 260-262: Diagnostics do not prevent recombination. Please redo this sentence.
line 276: remove "rabbit".
lines 317-321: this is too much detail compared to other sections. please remove this information.
line 332: "slightly shorter": please be more precise and include values/size.
lines 341-349: if this is new yam virus, this could be moved to section 5.
lines 364-365/lines 383-383: same information in different places? please check and edit accordingly.
section 3
Lines 450: please include which mealybug species?
lines 472-474: Is this information not provided yet? Some badnavirus groups are known to have endogenous sequences. Please redo.
lines 534-547: this section is confusing in the sense that it is the type of information that would go to a research paper and not a review paper? this section needs clarifications.
section 5: change to new yam viruses
I recommend manuscript for publication after the above suggestions are addressed.
Author Response
We thank Viruses for putting our paper through peer review. We believe the reviewers have done a great job in reviewing and the process has improved the quality of our paper. Attached please find the response to the second review. Thank you for reviewing our paper.
Regards
